# Thermodynamics of Duplication Thresholds in Synthetic Protocell Systems

**DOI:** 10.3390/life9010009

**Published:** 2019-01-15

**Authors:** Bernat Corominas-Murtra

**Affiliations:** Institute of Science and Technology Austria, Am Campus 1, A-3400 Klosterneuburg, Austria; bernat.corominas-murtra@ist.ac.at or bernatze@gmail.com

**Keywords:** protocell duplication, artificial life, thermodynamics of life, thermodynamics of duplication, stochastic thermodynamics

## Abstract

Understanding the thermodynamics of the duplication process is a fundamental step towards a comprehensive physical theory of biological systems. However, the immense complexity of real cells obscures the fundamental tensions between energy gradients and entropic contributions that underlie duplication. The study of synthetic, feasible systems reproducing part of the key ingredients of living entities but overcoming major sources of biological complexity is of great relevance to deepen the comprehension of the fundamental thermodynamic processes underlying life and its prevalence. In this paper an abstract—yet realistic—synthetic system made of small synthetic protocell aggregates is studied in detail. A fundamental relation between free energy and entropic gradients is derived for a general, non-equilibrium scenario, setting the thermodynamic conditions for the occurrence and prevalence of duplication phenomena. This relation sets explicitly how the energy gradients invested in creating and maintaining structural—and eventually, functional—elements of the system must always compensate the entropic gradients, whose contributions come from changes in the translational, configurational, and macrostate entropies, as well as from dissipation due to irreversible transitions. Work/energy relations are also derived, defining lower bounds on the energy required for the duplication event to take place. A specific example including real ternary emulsions is provided in order to grasp the orders of magnitude involved in the problem. It is found that the minimal work invested over the system to trigger a duplication event is around ~$10^{-13}\;J$, which results, in the case of duplication of all the vesicles contained in a liter of emulsion, in an amount of energy around ~1kJ. Without aiming to describe a truly biological process of duplication, this theoretical contribution seeks to explicitly define and identify the key actors that participate in it.

## 1. Introduction

How living beings have been able to overcome the entropic forces to develop increasingly complex individuals which, in turn, maintain their functionality is an open question and one of the hardest problems of modern science [1,2,3,4,5,6,7,8,9,10,11,12]. The exhaustive analysis of the energy flows in real living entities collides with the extreme complexity of even the simplest bacteria. Therefore, one must first set what are the *physical* defining properties of living beings and, then, try to attack the problem by cutting it into pieces. Each piece should incorporate a key or several key features, simple enough to accept a rigorous analysis, but complex enough to shed light to certain facets of the problem. The later integration of all pieces, however, will likely be much more than building a puzzle, for it is clear that the cross dependencies between all the building blocks will introduce an additional layer of complexity.

Following this philosophy, we will focus here on two crucial properties of living beings, according to accepted definitions of life discussed among scholars [13,14,15,16,17]. Specifically, we will concentrate on systems able to: (i) Capture material resources and turn them into building blocks by the use of externally provided free energy—and eventually undergo a duplication cycle and (ii) Keep its components together and distinguish itself from the environment. It is assumed that the compartment contains the metabolic and information system—if any. Our simplified system, thus, will lack two crucial features of living beings, namely, (iii) To process and transmit inheritable information to progeny and (iv) To undergo Darwinian evolution through variation of the copied inheritable information and a successive selection of the better progeny. We will thus focus on the thermodynamic properties of the *duplication* process, and we will skip all the complexity arising from other phenomena. It is worth to recall here that this kind of approach, where the essential physics of the duplication problem is addressed has a long history, dating back to the late thirties of the 20th century, with the highly influential works of N. Rashevsky [1].

In contrast to the usual top-down approaches followed in biology, we will address this problem using a bottom-up approach. In such kind of approaches to life-related phenomena, physical building blocks and chemical processes are externally assembled and triggered, creating artificial, synthetic entities that mimic some of the crucial properties of living beings. Consistently, this approach has been named *Artificial Life* [16,18,19,20]. Artificial cells, or, *protocells* are usually composed by emulsions [21] made of mixtures of lipids, precursors, and water [15,16,19,20,22,23,24,25,26,27,28]. The foundation of this approach is based on three main starting points: First, it provides a framework where energy imbalances trigger the emergence of cell like aggregates [21], second, it is possible to externally drive simplified metabolic reactions [15,16,26,28], and, third, it uses the same type of building blocks—mainly lipids—that compose an important part of the structure of most of the living organisms [29]. Crucial to our aims, it is worth to remark a couple of recent results: First, numerical approaches have shown that duplication dynamics as a consequence of energy imbalances due to geometrical frustration is expected in those systems, if properly driven out of equilibrium [30]. Second, recent experiments succeeded **in** duplicating real artificial protocells through a specific oil-in-water droplet system with replicating information templates [31]. This result is certainly remarkable, but our approach does exclude the role of any information/replication dynamics. In doing so, we explore how far can we go by just taking into account general stability properties and energy imbalances to explain and characterize the duplication process. The work presented here runs in parallel to an interesting complementary approach taken in [32], where the kinetics involved in the duplication events of synthetic systems was studied in detail.

In this paper we will work with a generic emulsion system [21]. We will make use of the well understood free energy landscape of such systems, where the contributions coming from aggregate geometry and size have been long studied [21,33,34], as well as the non-trivial contributions of the entropic terms [35,36]. The impact of a changing energy landscape—which eventually can favour a duplication event—will be studied from a generic non-equilibrium situation making use of modern methods arising from the emerging field of *Stochastic Thermodynamics* [37,38,39,40,41,42]. Within this framework, the evolution of the system can be studied following the individual trajectories in the phase space and, importantly, exact relations between energy and work can be obtained, even in out of equilibrium cases. In addition, relations between energy, entropy and information arise naturally [43,44].

The remainder of the paper is organized as follows: In the next section, we describe the thermodynamics of the abstract emulsion system in detail. We derive its free energy landscape, Section 2.1, the equilibrium distributions, Section 2.2, and the detailed balance condition over transitions, Section 2.3. Next, in Section 2.4, we expose the generic protocol that drives the system towards the occurrence of a duplication event. We end the section where the system is presented by exploring the orders of magnitude involved in these kind of systems, Section 2.5. Here we analyze the quantitative values of the thermodynamic functionals presented generically in the previous sections for a real microemulsion system. The thermodynamic analysis of the duplication thresholds is the core of section III. First, we derive a general relation for duplication probabilities, Section 3.1. Then, in Section 3.2, we explore the consequences of this result for a system evolving in a quasi-static fashion. Section 3.3 generalizes the previous equilibrium approach by providing an exact equality between probabilities of duplication thresholds in a specific non-equilibrium scenario, in which the relaxation process that may eventually lead to duplication happens between two states which may not be in equilibrium. This equivalence leads us to define general duplication scenarios and derive the general conditions of duplication, as well as the amount of work invested over the system to trigger a duplication event, and the conditions for the perpetuation of the duplication cycle. Section 3.4 refers to the free energy/entropy relations for the perpetuation of the duplication cycle in time. The final section is devoted to discuss the implications of the presented results. The whole paper is aimed to be self-contained and details of the derivations are provided in the Appendix A to make it understandable to non-specialized audiences.

## 2. The System

Our system is conceived as being an abstract emulsion in a kind of reaction tank of volume $V_{\mathrm{syst}}$ connected to a heat reservoir at inverse temperature $\beta =\frac{1}{T}$—we set $k_{B}=1$. Let $\vec{X}=\left(X_{1},\ldots ,X_{L}\right)$, where $X_{i}$ is a specific kind of lipid species populating the system and $\vec{Y}=\left(Y_{1},\ldots ,Y_{P}\right)$, where $Y_{i}$ is a specific kind of precursor/surfactant species populating the system. Let ${\vec{X}}_{\mathrm{tot}}=\left(X_{1,\mathrm{tot}},\ldots ,X_{L,\mathrm{tot}}\right),{\vec{Y}}_{\mathrm{tot}}=\left(Y_{1,\mathrm{tot}},\ldots ,Y_{P,\mathrm{tot}}\right)$ the total amount of molecules of the different species of lipids and precursors that lie in aqueous solution inside our volume. We refer to ${\vec{X}}_{\mathrm{tot}},{\vec{Y}}_{\mathrm{tot}}$ as the *boundary conditions*. As we shall see, they may change in time, under the action of an external protocol.

Due to the hydrophobic/hydrophilic nature of the surfactant molecules, we assume that (part of them) tend to aggregate in spheroidal compartments. Surfactants are supposed to populate the surface of the aggregates. No assumptions are made on the specific nature of the membranes or the interior of the aggregates, leaving the discussion always in a general plane. A *state*
$\sigma_{n}$ of our system is described by a 3-tuple:$$\sigma_{n}\equiv \sigma_{n}\left({\vec{X}}_{a},{\vec{Y}}_{a},n\right)\;,$$
where ${\vec{X}}_{a}=\left(X_{1,a},\ldots ,X_{L,a}\right)$ and ${\vec{Y}}_{a}=\left(Y_{1,a},\ldots ,Y_{P,a}\right)$ are the amount of lipids and precursors forming aggregates, respectively, and *n* the number of aggregates present in the volume. In general, and if no confusion can arise, we refer to a given state as $\sigma_{n}$ instead of $\sigma_{n}\left({\vec{X}}_{a},{\vec{Y}}_{a},n\right)$ for notational simplicity. We keep the label subscript “${}_{n}$” accounting for the number of aggregates only for notational convenience. When we introduce time dependence, we write $\sigma_{n}^{t}\equiv \sigma_{n}\left({\vec{X}}_{a}\left(t\right),{\vec{Y}}_{a}\left(t\right),n\left(t\right)\right)$. Not all molecules will be part of the aggregates. Therefore, we must account for these molecules in bulk. Consistently, given a state $\sigma_{n}\left({\vec{X}}_{a},{\vec{Y}}_{a},n\right)$ occuring under the boundary conditions ${\vec{X}}_{\mathrm{tot}},{\vec{Y}}_{\mathrm{tot}}$, we wil have that ${\vec{X}}_{b}={\vec{X}}_{\mathrm{tot}}-{\vec{X}}_{a}=\left(X_{1,b},\ldots ,X_{L,b}\right)$ and ${\vec{Y}}_{b}={\vec{Y}}_{\mathrm{tot}}-{\vec{Y}}_{a}=\left(Y_{1,b},\ldots ,Y_{P,b}\right)$ are the amount of lipids and precursors in bulk, respectively.

A *macrostate* or *coarse-grained* state ${\tilde{\sigma }}_{n}$ is defined as the 4-tuple:$${\tilde{\sigma }}_{n}\equiv {\tilde{\sigma }}_{n}\left({\vec{X}}_{\mathrm{tot}},{\vec{Y}}_{\mathrm{tot}},n,p\left(\sigma_{n}|{\tilde{\sigma }}_{n}\right)\right)\;,$$
where $p(\sigma_{n}|{\tilde{\sigma }}_{n})$ is the probability distribution of finding $\sigma_{n}$ as a particular realization of this macrostate. This macrostate can be realized through any state containing ${\vec{X}}_{\mathrm{tot}},{\vec{Y}}_{\mathrm{tot}}$ and *n* protocellular aggregates following the distribution $p(\sigma_{n}|{\tilde{\sigma }}_{n})$. In case of time dependence we write ${\tilde{\sigma }}_{n}^{t}\equiv {\tilde{\sigma }}_{n}^{t}\left({\vec{X}}_{\mathrm{tot}}\left(t\right),{\vec{Y}}_{\mathrm{tot}}\left(t\right),n,p\left(\sigma_{n}^{t}|{\tilde{\sigma }}_{n}^{t}\right)\right)$.

### 2.1. Gibbs Free Energy Landscape

The thermodynamic landscape of our system is given by the Gibbs free energy of the state $\sigma_{n}$,
$$G\left(\sigma_{n}\right)\equiv G_{{\vec{X}}_{\mathrm{tot}},{\vec{Y}}_{\mathrm{tot}}}\left(\sigma_{n}\right)\;.$$

The Gibbs free energy is always defined over states of the system and depends on both the state $\sigma_{n}$ and the boundary conditions ${\vec{X}}_{\mathrm{tot}},{\vec{Y}}_{\mathrm{tot}}$. Therefore, the same state will have energy changes if the boundary conditions change. Each macrostate has a uniquely defined free energy functional. For notational simplicity, we drop the subscript ${}_{{\vec{X}}_{\mathrm{tot}},{\vec{Y}}_{\mathrm{tot}}}$, if no confusion arises.

The complex nature of these type of emulsions results in a free energy functional with several blocks, which we construct step by step. First, we focus on the free energy contribution of a single protocellular aggregate, containing $\vec{X}$ lipids, $\vec{Y}$, and precursors, $G_{a}$:(1)$$G_{a}\left(X_{i},Y_{i}\right)=\sum_{i\leq L}\Delta \mu_{X_{i}}X_{i}+\sum_{i\leq P}\Delta \mu_{Y_{i}}Y_{i}+G_{\mathrm{geo}}\;,$$
where $\Delta \mu_{X_{i}}$ and $\Delta \mu_{Y_{i}}$ are the changes in chemical potential when moving lipids and surfactants from bulk into the *i*-th aggregate, and $G_{\mathrm{geo}}$ is a geometric term expressing shape and surface contributions to the free energy of the aggregate. This geometric term accounts for the membrane properties of the system, and is computed according to the existence of a minimum energy configuration or *perfect* protocellular aggregate, which can be directly computed as the optimal packing from the knowledge to the sizes and geometries of the precursor molecules. The geometrical term thus reads:(2)$$G_{\mathrm{geo}}=\gamma A+\frac{\alpha }{A}+\kappa \oint_{A}{\left(H-H_{0}\right)}^{2}dA\;,$$
where γ is the surface tension, α the compressibility coefficient, and κ the elastic bending modulus of the lipid membrane. The integral is the second order expansion of the contribution of the Helfrisch Hamiltonian to the overall free energy, being *H* the curvature of the membrane—as a function of some coordinates parametrizing the membrane surface—of the current aggregate and $H_{0}$ the curvature of the perfect aggregate. The integral is computed over the whole area of the membrane, *A* [33,34].

Once we have properly characterized the free energies of a single aggregate, we proceed to construct the free energy of the whole state $\sigma_{n}$. The next task will be to compute the entropy for a system in the state $\sigma_{n}=\sigma_{n}\left({\vec{X}}_{a},{\vec{Y}}_{a},n\right)$ under the boundary conditions ${\vec{X}}_{\mathrm{tot}},{\vec{Y}}_{\mathrm{tot}}$. To compute the entropy of such state, we apply directly Boltzmann’s definition over the amount of configurations the state $\sigma_{n}$ can adopt, $\Omega_{{\vec{X}}_{\mathrm{tot}},{\vec{Y}}_{\mathrm{tot}}}\left(\sigma_{n}\right)$ [45]:$$S\left(\sigma_{n}\right)=\log\Omega_{{\vec{X}}_{\mathrm{tot}},{\vec{Y}}_{\mathrm{tot}}}\left(\sigma_{n}\right)\;.$$

Clearly, $S\left(\sigma_{n}\right)\equiv S_{{\vec{X}}_{\mathrm{tot}},{\vec{Y}}_{\mathrm{tot}}}\left(\sigma_{n}\right)$. However, we do not write this dependence explicitly for the sake of readability, if no confusion can arise. This entropic term has two contributions, the *translational* entropy and the *configurational* entropy. We start with the translational contribution. We consider that the system of *n* indistinguishable aggregates has 3n degrees of freedom and that each aggregate diffuse around within a volume $V_{\mathrm{syst}}=nV_{a}$ and that $\langle \ell_{m}\rangle$ is an appropriate length scale for such a diffusive process. Accordingly, one has that the amount of configurations provided by the translational term is:$$∼\frac{1}{n!}{\left(\frac{nV_{a}}{\langle \ell_{m}\rangle }\right)}^{n}\;.$$

We emphasize that, in the approach take here, $\langle \ell_{m}\rangle$ has been chosen as a typical volume unit whose purpose is to render the argument of the logarithm dimensionless—for a deeper discussion on the choice of the right length scale see [35,36]. For each configuration described above, we must account for the potential degeneracy of states, or, in other words, the amount of configurations given by the amount of molecules in bulk and forming the aggregates. For each chemical species, e.g., the *i*-th lipid, this amount of configurations is
$$∼\left(\begin{array}{c} X_{i,\mathrm{tot}}\\ X_{i,a} \end{array}\right)\;.$$

Therefore, assuming that there are no cross dependencies among the different chemical species, one has that the amount of configurations of molecules in bulk and aggregates is: $$∼\prod_{i\leq L}\prod_{k\leq P}\left(\begin{array}{c} X_{i,\mathrm{tot}}\\ X_{i,a} \end{array}\right)\left(\begin{array}{c} Y_{i,\mathrm{tot}}\\ Y_{i,a} \end{array}\right)\;.$$

Considering these two contributions, the entropy term reads:$$S\left(\sigma_{n}\right)=\log\left[\frac{1}{n!}{\left(\frac{nV_{a}}{\langle v_{m}\rangle }\right)}^{n}\prod_{i\leq L}\prod_{k\leq P}\left(\begin{array}{c} X_{i,\mathrm{tot}}\\ X_{i,a} \end{array}\right)\left(\begin{array}{c} Y_{i,\mathrm{tot}}\\ Y_{i,a} \end{array}\right)\right]\;.$$

The overall entropy of the state $\sigma_{n}=\sigma_{n}\left({\vec{X}}_{a},{\vec{Y}}_{a},n\right)$, under the boundary conditions given by ${\vec{X}}_{\mathrm{tot}},{\vec{Y}}_{\mathrm{tot}}$, $S\left(\sigma_{n}\right)=\log\Omega_{{\vec{X}}_{\mathrm{tot}},{\vec{Y}}_{\mathrm{tot}}}\left(\sigma_{n}\right)$, is:(3)$$\begin{array}{ccc} S(\sigma_{n}) & = & n\log\left(\frac{V_{a}}{\langle v_{m}\rangle }\right)+\\ & & +\sum_{i\leq L}\log\left(\begin{array}{c} X_{i,\mathrm{tot}}\\ X_{i,a} \end{array}\right)+\sum_{i\leq P}\log\left(\begin{array}{c} Y_{i,\mathrm{tot}}\\ Y_{i,a} \end{array}\right)\;, \end{array}$$
where we used the fact the log(ab)=loga+logb and the Stirling approximation for the factorial for the first term, namely logn!≈nlogn−n. Collecting all the above ingredients, we have that the Gibbs free energy of the system in the state $\sigma_{n}=\sigma_{n}\left({\vec{X}}_{a},{\vec{Y}}_{a},n\right)$ under boundary conditions ${\vec{X}}_{\mathrm{tot}},{\vec{Y}}_{\mathrm{tot}}$ becomes:(4)$$\begin{array}{ccc} G(\sigma_{n}) & = & \sum_{i\leq L}\mu_{X_{i}}^{\circ }X_{i,\mathrm{tot}}+\sum_{i\leq P}\mu_{Y_{i}}^{\circ }Y_{i,\mathrm{tot}}+\\ & & \;\;\;\;+\sum_{i\leq n}G_{a}\left(X_{i},Y_{i}\right)-TS\left(\sigma_{n}\right)\;, \end{array}$$
with the standard chemical potentials $\mu_{X_{i}}^{\circ }$ and $\mu_{Y_{i}}^{\circ }$ of lipids and precursors, respectively.

### 2.2. Helmholtz Free Energy

Let the system be subject to the boundary conditions ${\vec{X}}_{\mathrm{tot}},{\vec{Y}}_{\mathrm{tot}}$. In equilibrium, the probability that the system is in the particular state $\sigma_{n}$, belonging to the macrostate ${\tilde{\sigma }}_{n}$ is given by the Boltzmann distribution, $p(\sigma_{n}|{\tilde{\sigma }}_{n})$ [45]:(5)$$p\left(\sigma_{n}|{\tilde{\sigma }}_{n}\right)=\frac{e^{-\beta G(\sigma_{n})}}{Z({\tilde{\sigma }}_{n})}\;,$$
being $Z({\tilde{\sigma }}_{n})$ the partition function, namely:(6)$$Z\left({\tilde{\sigma }}_{n}\right)=\sum_{\sigma_{n}\in {\tilde{\sigma }}_{n}}e^{-\beta G(\sigma_{n})}\;.$$

Accordingly, the Helmholtz free energy of the macrostate ${\tilde{\sigma }}_{n}$, $F({\tilde{\sigma }}_{n})$ is:(7)$$\begin{array}{c} F\left({\tilde{\sigma }}_{n}\right)=-\log Z\left({\tilde{\sigma }}_{n}\right)={\langle G\rangle }_{{\tilde{\sigma }}_{n}}-\frac{1}{\beta }H\left({\tilde{\sigma }}_{n}\right)\;, \end{array}$$
being ${\langle \ldots \rangle }_{{\tilde{\sigma }}_{n}}$ the average over all states of the macrostate and $H({\tilde{\sigma }}_{n})$ the entropy of the macrostate, namely:$$H\left({\tilde{\sigma }}_{n}\right)=-\sum_{\sigma_{n}\in {\tilde{\sigma }}_{n}}p\left(\sigma_{n}|{\tilde{\sigma }}_{n}\right)\log p\left(\sigma_{n}|{\tilde{\sigma }}_{n}\right)\;,$$
where $p(\sigma_{n}|\tilde{\sigma_{n}})$ is now defined as:$$p\left(\sigma_{n}|{\tilde{\sigma }}_{n}\right)=e^{-\beta (G\left(\sigma_{n}\right)+F\left({\tilde{\sigma }}_{n}\right))}\;.$$

We point out that we will refer to a given probability distribution associated to a macrostate ${\tilde{\sigma }}_{n}$ either as $p(\sigma_{n}|{\tilde{\sigma }}_{n})$ or $p_{|{\tilde{\sigma }}_{n}}$, indistinctly. We finally recall that we assume that the equilibrium distribution macrostate ${\tilde{\sigma }}_{n}$ is such that:$$\underset{\sigma_{n}\in {\tilde{\sigma }}_{n}}{\mathrm{argmin}}\left\{G_{{\vec{X}}_{\mathrm{tot}},{\vec{Y}}_{\mathrm{tot}}}\left(\sigma_{n}\right)\right\}=\underset{\sigma }{\mathrm{argmin}}\left\{G_{{\vec{X}}_{\mathrm{tot}},{\vec{Y}}_{\mathrm{tot}}}\left(\sigma \right)\right\}\;,$$
where we emphasized the dependency on the boundary conditions ${\vec{X}}_{\mathrm{tot}},{\vec{Y}}_{\mathrm{tot}}$ only for clarity. In words, we assume that the equilibrium distribution is defined around the absolute minimum of Gibbs free energies, and that such a minimum is unique.

### 2.3. Detailed Balance Condition in Duplication

The process of duplication/fusion of aggregates is of special interest for us, since it is the basis of duplication. It is assumed to satisfy the following transition rates between states:$$\begin{array}{ccc} \sigma_{n}\left({\vec{X}}_{a},{\vec{Y}}_{a},n\right) & \overset{k_{n}^{+}n}{\to } & \sigma_{n+1}\left({\vec{X}}_{a},{\vec{Y}}_{a},n+1\right)\\ \sigma_{n}\left({\vec{X}}_{a},{\vec{Y}}_{a},n\right) & \overset{k_{n}^{-}n}{\to } & \sigma_{n-1}\left({\vec{X}}_{a},{\vec{Y}}_{a},n-1\right)\;, \end{array}$$
where the kinetic constants relate as:(8)$$k_{n}^{-}=k_{n}^{+}e^{-\beta \delta G(\sigma_{n},\sigma_{n+1})}\;,$$
where $\delta G\left(\sigma_{n},\sigma_{n+1}\right)\equiv G\left(\sigma_{n+1}\right)-G\left(\sigma_{n}\right)$. Detailed balance condition is also assumed for any other transition between states. Therefore, for any two states $\sigma_{n}\in {\tilde{\sigma }}_{n}$ and $\sigma_{n+1}\in {\tilde{\sigma }}_{n+1}$, thanks to the detailed balance condition given in Equation (Equation 8) and assumed for all transitions, one has that, between two arbitrary states σ, $\sigma^{\prime }$:(9)$$\frac{p(\sigma^{\prime }\to \sigma )}{p(\sigma \to \sigma^{\prime })}\approx \frac{k_{n}^{-}}{k_{n}^{+}}=e^{-\beta \delta G(\sigma ,\sigma^{\prime })}\;.$$

Importantly, we recall that the functional *G* must be computed under the same boundary conditions ${\vec{X}}_{\mathrm{tot}},{\vec{Y}}_{\mathrm{tot}}$ in any evaluation of the difference, i.e.,:$$\delta G\left(\sigma ,\sigma^{\prime }\right)=G\left(\sigma^{\prime }\right)-G\left(\sigma \right)\equiv G_{{\vec{X}}_{\mathrm{tot}},{\vec{Y}}_{\mathrm{tot}}}\left(\sigma^{\prime }\right)-G_{{\vec{X}}_{\mathrm{tot}},{\vec{Y}}_{\mathrm{tot}}}\left(\sigma \right)\;.$$

### 2.4. The Driving Protocol

Let us assume that at time t=0 the system is in contact to a thermal reservoir at inverse temperature β, and in an equilibrium macrostate ${\tilde{\sigma }}_{n}^{0}$, that is—see Figure 1):$$p\left(\sigma_{n}^{0}|{\tilde{\sigma }}_{n}^{0}\right)=\frac{1}{Z({\tilde{\sigma }}_{n}^{0})}e^{-\beta G(\sigma_{n}^{0})}\;.$$

From this moment on, we run a protocol that changes the energy landscape, without separating the system from the heath bath neither changing the whole system’s volume, $V_{\mathrm{syst}}$. This protocol runs from t=0 to t=τ—see Figure 1b,c. For example, suppose that we add new lipids and that we switch on a light that triggers a metabolic reaction that transforms lipids into precursors, thereby creating new surfactants. We call this protocol ψ(t). In general, it will affect the L+P variables of our system. Therefore, the protocol ψ(t) consists on a list of—maybe interdependent—protocols:$$\psi \left(t\right)=(\phi_{1}\left(t\right),\ldots ,\phi_{L}\left(t\right),\phi_{L+1}\left(t\right),\ldots ,\phi_{L+P}\left(t\right))\;,$$
where the first *L* elements $\phi_{1}\left(t\right),\ldots ,\phi_{L}\left(t\right)$ explicit the action of the protocol on the lipids $X_{1},\ldots ,X_{L}$ abundance and the last *P* elements $\phi_{L+1}\left(t\right),\ldots ,\phi_{L+P}\left(t\right)$ explicit the change due to the protocol on the precursors $Y_{1},\ldots ,X_{P}$ abundance. Let us be more specific on the action of the protocol. Assume that at time *t* the boundary conditions of our system are given by ${\vec{X}}_{\mathrm{tot}}\left(t\right),{\vec{Y}}_{\mathrm{tot}}\left(t\right)$. The application of the protocol a for short time interval [t,t+δ] to the boundary conditions, denoted by $\delta \psi ({\vec{X}}_{\mathrm{tot}}\left(t\right),{\vec{Y}}_{\mathrm{tot}}\left(t\right))$ will lead the boundary conditions to change as:$$\begin{array}{ccc} \delta \psi ({\vec{X}}_{\mathrm{tot}}\left(t\right),{\vec{Y}}_{\mathrm{tot}}\left(t\right)) & = & (X_{1}\left(t\right)+\delta \phi_{1}\left(t\right),\ldots \\ & & \ldots ,Y_{1}\left(t\right)+\delta \phi_{L+1}\left(t\right)\phi_{L+1}\left(t\right),\ldots )\;. \end{array}$$

The above transformation of the boundary conditions will lead the system to change its macrostate, from ${\tilde{\sigma }}^{t}$ to ${\tilde{\sigma }}^{t+\delta }$. This transition can be done through a set of stochastic trajectories, which will be referred to as Σ[t,t+δ]. At τ the system will be at the macrostate ${\tilde{\sigma }}_{n+1}^{\tau }$ and we will stop the protocol—see Figure 1d—letting the system relax towards an equilibrium state, achieved at time $\tau_{\infty }$—see Figure 1d. The distribution of states $p_{|{\tilde{\sigma }}_{n+1}^{\tau_{\infty }}}$ is assumed to obey the standard equilibrium Boltzmann statistics:$$p\left(\sigma_{n+1}^{\tau_{\infty }}|{\tilde{\sigma }}_{n+1}^{\tau_{\infty }}\right)=\frac{1}{Z({\tilde{\sigma }}_{n+1}^{\tau_{\infty }})}e^{-\beta G(\sigma_{n+1}^{\tau_{\infty }})}\;.$$

We assume that a duplication event has taken place in the time interval [τ−δ,τ] and that the relaxation process happening at the interval $\left[\tau ,\tau_{\infty }\right]$ does not imply a change in the number of protocell aggregates. We remind that the whole process takes place in contact to a heat reservoir with inverse temperature β and at a constant volume $V_{\mathrm{syst}}$.

### 2.5. An Example: Ternary Emulsions

To grasp the orders of magnitude involved in our problem, we take a particular example of the above general system, in line to the one described in [26,27]. From this example, we perform a rough estimation of the orders of magnitude involved in the computation of the free energies of a single aggregate. For the sake of readability and extension, the computations provided here are not as detailed as in the other parts of the paper. We refer the interested reader to [21,26,27,46,47,48] for the detailed discussions on the orders of magnitude and potential experimental set ups.

Suppose that we have a Winsor type IV ternary emulsion made of a single lipid, X≡
*decanoic acid anhydride*, ($C_{9}H_{19}-\mathrm{CO}-O-\mathrm{OC}-C_{9}H_{19}$), a single precursor, Y≡
*decanoic acid*, ($C_{9}H_{19}-\mathrm{COOH}$), and water. Equation (Equation 1) now reads:$$G_{a}\left(X,Y\right)=\Delta \mu_{X}X+\Delta \mu_{Y}Y+G_{\mathrm{geo}}\;.$$
$\Delta \mu_{Y}$ can be calculated from their partition coefficient—i.e., the fraction of lipids found in bulk solution as opposed to the aggregates. Estimations give this value to be around 14% [46]. If $k_{Y}^{+}/k_{Y}^{-}$ is the ratio between precursor molecules going from bulk to aggregates and precursor molecules going from aggregates to bulk, this reads:$$\frac{k_{Y}^{+}}{k_{Y}^{-}}\approx 0.14\;.$$

Therefore, using Equation (Equation 8), and setting $\beta =1/k_{B}T$, one can approximate the energy gain of moving a decanoic acid molecule from bulk to aggregate, $\Delta \mu_{Y}$, as
$$\Delta \mu_{Y}\approx \log\left(0.14\right)k_{B}T\;,$$
where $k_{B}$ is the Boltzmann constant, $k_{B}\approx 1.38\times 10^{-23}\;J/K$ At T=300K, and $N_{A}$ being the Avogadro number, the above equation leads to:$$N_{A}\Delta \mu_{Y}\approx -4.9\;\mathrm{kJ}/\mathrm{mol}\;.$$

Since the decanoic acid anhydride ($C_{9}H_{19}-\mathrm{CO}-O-\mathrm{OC}-C_{9}H_{19}$) has two hydrophobic chains, we set
$$\Delta \mu_{X}=2\Delta \mu_{Y}\approx -9.8\;\mathrm{kJ}/\mathrm{mol}\;,$$
which in turn evaluates to a partition coefficient of ∼2%. For the geometric term given by Equation (Equation 2), we make the assumption that γ,α≫κ, therefore the contribution of the Helfrisch hamiltonian will not be taken into account. The surface tension and the compressibility parameters, γ,α can be estimated as γ≈45.9mN/m and $\alpha \approx 5.80\times 10^{-45}\;{\mathrm{Nm}}^{3}$ [26]. We assume a spherical lipid core of $X_{c}$ precursor molecules, whose individual molecular volume $V_{X}=0.54\;{\mathrm{nm}}^{3}$. Thus, the spherical core of the aggregate has a radius, $R_{\mathrm{core}}\left(X_{c}\right)$, of:$$R_{\mathrm{core}}\left(X_{c}\right)∼{\left(\frac{3V_{X}}{4\pi }X_{c}\right)}^{1/3}\;.$$

The whole aggregate, including the surface molecules, displays a radius, $R_{\circ }\left(X_{c}\right)$, of
$$R_{\circ }\left(X_{c}\right)∼R_{\mathrm{core}}\left(X_{c}\right)+\ell_{t}\;,$$
where $\ell_{t}$ is the length of the tail of the surfactant molecules, which is considered constant. The optimal number of surfactant molecules, $Y^{★}\left(X_{c}\right)$ for this amount of molecules in the core of the aggregate is then computed as:(10)$$Y^{★}\left(X_{c}\right)=\frac{4\pi }{a_{0}}R_{\circ }^{2}\left(X_{c}\right)\;.$$

We assume that the tail length of the surfactants is around $\ell_{t}=1.4\;\mathrm{nm}$ and that their effective head area $a_{0}=25\;{Å}^{2}$ [21]. The typical radius of oil droplets is around 100nm leading to a volume of $\approx 4.1\times 10^{6}$ nm${}^{3}$, i.e., ~0.004 femtoliter, which—assuming a typical water-to-oil ratio of 10:1—gives a system volume of 0.044 femtoliter per droplet. Therefore, a milliliter of emulsion has an order of magnitude of $10^{13}$ oil droplets. From the ratio of precursor to droplet volume, it follows that $X_{c}\approx 7.62\times 10^{6}$. With an optimal packing number of surfactants $Y^{★}\left(X_{c}\right)$ computed from Equation (Equation 10) and a partition coefficient of 14%, one can estimate a total of $Y_{c}=5.7\times 10^{5}$ surfactant molecules. With these values, a rough estimation of the orders of magnitude of the free energy of a single aggregate whose packing is optimal, $G_{a}^{★}\left(X,Y\right)$, is given by:(11)$$|G_{a}^{★}\left(X,Y\right)|∼10^{-13}J\;.$$

This example gives us an orientation about the energy scales involved in our problem.

## 3. Duplication Thresholds

We proceed now to explore under which circumstances the application of the protocol results into a duplication event. The goal is to obtain an inequality which, when observed, a duplication event is expected to take place. This will be related to the amount of work performed from the protocol. We perform the analysis from a quasi-static approach and from a more general non-equilibrium approach. First of all, we derive a general condition for the transition probabilities among macrostates, which does not require, a priori, equilibrium conditions.

### 3.1. Transition Probabilities between Macrostates

Now we take a close analysis on the process happening in the interval [t,t+δ], where 0<t<τ. We drop the indices ${}_{n}$ because, we consider transitions between any two states, and, by now, there will not be necessarily a duplication event in consideration. At time *t* the Gibbs free energy landscape suffers a change imposed by the protocol ψ(t). The initial state, ${\tilde{\sigma }}^{t}$, is therefore perturbed and may no longer be necessarily in equilibrium. The system then relaxes to ${\tilde{\sigma }}^{t+\delta }$, which may not be in equilibrium, too. The boundary conditions $\left({\vec{X}}_{\mathrm{tot}},{\vec{Y}}_{\mathrm{tot}}\right)$ are considered constant after the change imposed by the protocol at time *t* until time t+δ. The probability of jumping from macrostate ${\tilde{\sigma }}^{t}$ to macrostate ${\tilde{\sigma }}^{t+\delta }$ is given by:$$\begin{array}{c} p\left({\tilde{\sigma }}^{t}\to {\tilde{\sigma }}^{t+\delta }\right)=\sum_{\sigma^{t}}\sum_{\sigma^{t+\delta }}p\left(\sigma^{t}|{\tilde{\sigma }}^{t}\right)p\left(\sigma^{t}\to \sigma^{t+\delta }\right)\;. \end{array}$$

Thanks to the detailed balance condition, one can rewrite the backwards transition as:$$\begin{array}{c} p\left({\tilde{\sigma }}^{t+\delta }\to {\tilde{\sigma }}^{t}\right)=\sum_{\sigma^{t}}\sum_{\sigma^{t+\delta }}p\left(\sigma^{t}|{\tilde{\sigma }}^{t}\right)p\left(\sigma^{t}\to \sigma^{t+\delta }\right)g\left(t,t+\delta \right)\;, \end{array}$$
where g(t,t+δ) is a function that depends on the states, $\sigma^{t},\sigma^{t+\delta }$ which, written in a suitable form for further developments, reads:$$g\left(t,t+\delta \right)\equiv \frac{e^{\beta \delta G(t,t+\delta )}}{e^{\ln\frac{p(\sigma^{t}|{\tilde{\sigma }}^{t})}{p(\sigma^{t+\delta }|{\tilde{\sigma }}^{t+\delta })}}}\;.$$

Now, we derive the probability that we chose a given trajectory $\sigma_{n}^{\tau -\delta }$ to $\sigma_{n+1}^{\tau }$ from the ensemble Σ[t,t+δ] of trajectories that go from ${\tilde{\sigma }}^{t}$ to ${\tilde{\sigma }}^{t+\delta }$—see Figure 2. This probability distribution is referred to as $p_{\to }^{t+\delta }$, and is defined as:(12)$$p_{\to }^{t+\delta }\left(\sigma^{t},\sigma^{t+\delta }\right)\equiv \frac{p\left(\sigma^{t}|{\tilde{\sigma }}^{t+\delta }\right)p\left(\sigma^{t}\to \sigma^{t+\delta }\right)}{p({\tilde{\sigma }}^{t}\to {\tilde{\sigma }}^{t+\delta })}\;.$$

Conversely, we can define the backwards version of the above probability distribution, namely, $p_{\leftarrow }^{t+\delta }$ as:(13)$$p_{\leftarrow }^{t+\delta }\left(\sigma^{t+\delta },\sigma^{t}\right)\equiv \frac{p\left(\sigma^{t+\delta }|{\tilde{\sigma }}^{t+\delta }\right)p\left(\sigma^{t+\delta }\to \sigma^{t}\right)}{p({\tilde{\sigma }}^{t+\delta }\to {\tilde{\sigma }}^{t})}\;.$$

The probability $p_{\leftarrow }^{t+\delta }$ accounts for the probability of a given trajectory in case of time reversal of the protocol action. It is straightforward to check that both $p_{\to }^{t+\delta }$ and $p_{\leftarrow }^{t+\delta }$ are well defined probability distributions—i.e., that they sum up to 1. With the above defined distributions, the above computations lead to:$$\begin{array}{c} \frac{p({\tilde{\sigma }}^{t+\delta }\to {\tilde{\sigma }}^{t})}{p({\tilde{\sigma }}^{t}\to {\tilde{\sigma }}^{t+\delta })}=\sum_{\sigma^{t}}\sum_{\sigma^{t+\delta }}p_{\to }^{t+\delta }\left(\sigma^{t},\sigma^{t+\delta }\right)g\left(t,t+\delta \right)\;. \end{array}$$

The above equation is the average over all paths of the last element of the sum, namely: (14)$$\frac{p({\tilde{\sigma }}^{t+\delta }\to {\tilde{\sigma }}^{t})}{p({\tilde{\sigma }}^{t}\to {\tilde{\sigma }}^{t+\delta })}={\langle \frac{e^{\beta \delta G(t,t+\delta )}}{e^{\ln\frac{p(\sigma^{t}|{\tilde{\sigma }}^{t})}{p(\sigma^{t+\delta }|{\tilde{\sigma }}^{t+\delta })}}}\rangle }_{\Sigma [t,t+\delta ]}\;,$$
where the brackets 〈…〉 denote average over all the microscopic trajectories Σ[t,t+δ] between states $\sigma^{t}\to \sigma^{t+\delta }$ that realize the transition from macrostate ${\tilde{\sigma }}^{t}$ to macrostate ${\tilde{\sigma }}^{t+\delta }$.

### 3.2. Quasi-Static Approach

The first exploration corresponds to the situation in which the transitions triggered by the protocol ψ(t) are performed quasistatically, that is: They are so slow that all the trajectories Σ[0,τ] can be considered a succession of equilibrium states. Applying the general relation given by Equation (Equation 14) we arrive, after cancellations, at:$$\frac{p({\tilde{\sigma }}^{t}\to {\tilde{\sigma }}^{t+\delta })}{p({\tilde{\sigma }}^{t+\delta }\to {\tilde{\sigma }}^{t})}=\frac{Z({\tilde{\sigma }}^{t+\delta })}{Z({\tilde{\sigma }}^{t})}\;,$$
and we then recover, as expected, the equilibrium relation for the backwards and forwards probabilities:(15)$$\frac{p({\tilde{\sigma }}^{t}\to {\tilde{\sigma }}^{t+\delta })}{p({\tilde{\sigma }}^{t+\delta }\to {\tilde{\sigma }}^{t})}=e^{-\beta \delta F(t,t+\delta )}\;,$$
where δF(t,t+δ) is the increase of the Helmholtz free energies—see Equation (Equation 7)—through the interval [t,t+δ]:$$\delta F\left(t,t+\delta \right)=F\left({\tilde{\sigma }}^{t+\delta }\right)-F\left({\tilde{\sigma }}^{t}\right)\;.$$

As stated in the description of the protocol ψ(t), we assume that a duplication event has taken place in the time interval [τ−δ,τ]. To study this case, we recover the subscripts ${}_{n}$, ${}_{n+1}$ accounting for the number of aggregates in the system. This will imply that, at time τ−δ, we had the system in a macrostate ${\tilde{\sigma }}_{n}^{\tau -\delta }$ and that at time τ the system transitioned towards a macrostate state ${\tilde{\sigma }}_{n+1}^{\tau }$. For that to happen spontaneously, we need that:$$p\left({\tilde{\sigma }}_{n}^{\tau -\delta }\to {\tilde{\sigma }}_{n+1}^{\tau }\right)>p\left({\tilde{\sigma }}_{n+1}^{\tau }\to {\tilde{\sigma }}_{n}^{\tau -\delta }\right)\;.$$
and, according to Equation (Equation 15), one needs that δF(t,t+δ)<0, which implies:$$F\left({\tilde{\sigma }}_{n+1}^{\tau }\right)<F\left({\tilde{\sigma }}_{n}^{\tau -\delta }\right)\;.$$

If we take a closer look to the structure of the Gibbs free energy given in Equation (Equation 4), we can refine the above condition. Indeed, since the boundary conditions ${\vec{X}}_{\mathrm{tot}},{\vec{Y}}_{\mathrm{tot}}$ do not change during the time interval [τ−δ,τ), the contributions to the change of the free energies will only correspond to the free energies of the aggregates, due to size and frustration given in Equation (Equation 1), and their associated entropic terms, given by the Shannon entropies of the macrostate and the translational and configurational entropies of the states, given in Equation (Equation 3). After cancellations, we arrive to:(16)$$\begin{array}{c} \beta \delta \langle G_{a}\rangle <\delta S\left(\tau -\delta ,\tau \right)\;, \end{array}$$
where the increase on the average free energies $\delta \langle G_{a}\rangle$ is defined from Equation (Equation 1) as:(17)$$\delta \langle G_{a}\rangle \equiv {\langle \sum_{i\leq n+1}G_{a}\left(X_{i},Y_{i}\right)\rangle }_{{\tilde{\sigma }}_{n+1}^{\tau }}-{\langle \sum_{i\leq n}G_{a}\left(X_{i},Y_{i}\right)\rangle }_{{\tilde{\sigma }}_{n}^{\tau -\delta }},$$
being the averages taken over the whole set of states belonging to ${\tilde{\sigma }}_{n+1}^{\tau }$ and ${\tilde{\sigma }}_{n}^{\tau -\delta }$, respectively, and the entropic gradient δS(τ−δ,τ) is defined as:(18)δS(τ−δ,τ)=δS(τ−δ,τ)+δH(τ−δ,τ).
where δS(τ−δ,τ) is the increase of configurational and translational entropies for each state, as given in Equation (Equation 3):$$\delta S\left(\tau -\delta ,\tau \right)={\langle S\left(\sigma_{n+1}^{\tau }\right)\rangle }_{{\tilde{\sigma }}_{n+1}^{\tau }}-{\langle S\left(\sigma_{n}^{\tau -\delta }\right)\rangle }_{{\tilde{\sigma }}_{n}^{\tau -\delta }}\;.$$

And δH(τ−δ,τ) the increase on Shannon entropies, namely:$$\delta H\left(\tau -\delta ,\tau \right)=H\left({\tilde{\sigma }}_{n+1}^{\tau }\right)-H\left({\tilde{\sigma }}_{n}^{\tau -\delta }\right)\;.$$

Knowing the evolution of free energies, and assuming the quasi-static approach, one can easily compute the amount of work performed by the protocol ψ(t) to trigger a duplication event. Indeed, in the quasi static approach, the amount of work δw(τ−δ,τ) invested over the system can be identified with the Helmholtz free energy gradients, namely:δw(τ−δ,τ)=δF(τ−δ,τ).

In consequence, the amount of work performed over the system along the protocol, $W_{\psi }$ is, assuming a continuous approach (δF(t,t+δ)→dF(t)):$$W_{\psi }=\int_{0}^{\tau }dF\left(t\right)=\Delta F\left(0,\tau \right)\;,$$
as expected in the case of equilibrium transformations [49].

### 3.3. Non Equilibrium Approach

We now explore a more general situation, in which the states visited along the trajectory are not necessarily in equilibrium, and thus, an extra amount of dissipated heat is expected, deforming the energy/work relations derived in the previous section [37,38,50]—see Figure 3. Our approach does not consider explicitly other sources of non-equilibrium behaviour, and is focused on the exploration of the potentials under the assumption that the final and initial states may not be equilibrium ones. In particular, the hypothesis of detailed balance between different states of the system is always assumed to hold at the level of microscopic transitions.

Specifically, let us consider the case in which at time *t* the boundary conditions ${\vec{X}}_{\mathrm{tot}},{\vec{Y}}_{\mathrm{tot}},$ suffer a sudden change imposed by the protocol ψ(t). We observe that the change induced by the protocol to the boundary conditions implies a change on the free energy landscape described by the Gibbs free energy in Equation (Equation 4). The initial state, ${\tilde{\sigma }}^{t}$, is therefore perturbed and is not necessarily considered in equilibrium. We consider that, in this irreversible destabilization of the system, an amount of entropy like:$$∼\beta Q_{\psi }\left(t\right)\;,$$
is produced, due to the non-equilibrium transformation that is associated to the perturbation of the system after the application of the protocol. This part will not be studied in detail, since it plays no role in the duplication process. The system then moves to ${\tilde{\sigma }}^{t+\delta }$, which may not be in equilibrium, too. As above, the boundary conditions $\left({\vec{X}}_{\mathrm{tot}},{\vec{Y}}_{\mathrm{tot}}\right)$ are considered constant in the interval [t,t+δ] after the change imposed by the protocol at time *t*.

If we do not assume a priori that the starting distribution $p(\sigma_{n}^{t}|{\tilde{\sigma }}^{t})$ is an equilibrium one, we see that, under the application of Equation (Equation 14) we reach a more general relation—see Appendix A for details:(19)$$\frac{p({\tilde{\sigma }}^{t}\to {\tilde{\sigma }}^{t+\delta }))}{p({\tilde{\sigma }}^{t+\delta }\to {\tilde{\sigma }}^{t})}\geq e^{-\beta ({\langle \delta G\left(t,t+\delta \right)\rangle }_{\Sigma [t,t+\delta ]})+\delta H\left(t,t+\delta \right)}\;,$$
where ${\langle \delta G\left(t,t+\delta \right)\rangle }_{\Sigma [t,t+\delta ]}$ is the increase of Gibbs free energy averaged over all trajectories Σ[t,t+δ] from ${\tilde{\sigma }}^{t}$ to ${\tilde{\sigma }}^{t+\delta }$. Unfortunately, the above inequality only can give necessary but not sufficient conditions for duplication. The derivation of an exact equivalence for a restricted range of situations—yet involving many non-equilibrium cases—is the objective of the next subsection.

#### 3.3.1. Free Energy Structure

To achieve an exact relation between forwards and backwards probabilities of duplication, we need to develop some equivalences involving information-theoretic measures. These relations are derived from the exploration of the structure of 〈δG(t,t+δ)〉. It is important to highlight that, since we do not assume we are in equilibrium, one cannot use the identity $F\left({\tilde{\sigma }}^{t}\right)={\langle G\rangle }_{{\tilde{\sigma }}^{t}}-\beta^{-1}H\left({\tilde{\sigma }}^{t}\right)$ anymore. Again, we focus our efforts in the study of the time interval [τ−δ,τ], where we assume that a duplication event has taken place. As above, we recover the subscripts ${}_{n}$, ${}_{n+1}$ accounting for the number of aggregates in the system. We remind that this implies that, at time τ−δ we had the system in a macrostate state ${\tilde{\sigma }}_{n}^{\tau -\delta }$ and that at time τ the system transitioned towards a macrostate state ${\tilde{\sigma }}_{n+1}^{\tau }$.

**First relation**. Observe that we can decouple the general term ${\langle \delta G\left(\tau -\delta ,\tau \right)\rangle }_{\Sigma [\tau -\delta ,\tau ]}$ as follows: Let $p_{|{\tilde{\sigma }}^{t}}$ be the probability distribution of the actual macrostate ${\tilde{\sigma }}^{t}$, and $p_{|{\tilde{\sigma }}_{*}^{t}}$ be the equilibrium distribution *associated to the equilibrium macrostate*
${\tilde{\sigma }}^{t}$, under the conditions imposed by the protocol at time *t*. That is, the probability distribution that would correspond to ${\tilde{\sigma }}^{t}$ if it where in equilibrium, ${\tilde{\sigma }}^{t}={\tilde{\sigma }}_{*}^{t}$. In other words, we have an equilibrium distribution $p\left(\sigma^{t}|{\tilde{\sigma }}_{*}^{t}\right)∼e^{-\beta G(\sigma^{t})}$, sharing the *support* (let *p* be a probability distribution defined over the set $X^{\prime }$ and let $X\subseteq X^{\prime }$ such that $X=\{x_{i}\in X^{\prime }:p\left(x_{i}\right)>0\}$. *X* is the *support set* of the probability distribution *p*. In words, the support set is the set of elements whose probability is larger than 0) set with the actual, possibly non-equilibrium, distribution $p_{|{\tilde{\sigma }}^{t}}$. After rearrangements one finds that—see Appendix A for details:(20)$${\langle \beta G\left(\tau -\delta \right)\rangle }_{{\tilde{\sigma }}_{n}^{\tau -\delta }}=H\left({\tilde{\sigma }}_{n}^{\tau -\delta }\right)+D(p_{|{\tilde{\sigma }}_{n}^{\tau -\delta }}\left|\right|p_{|{\tilde{\sigma }}_{n*}^{\tau -\delta }})+F\left({\tilde{\sigma }}_{n*}^{t}\right),$$
where $D(p_{|{\tilde{\sigma }}_{n}^{\tau -\delta }}||p_{|{\tilde{\sigma }}_{n*}^{\tau -\delta }})$ is the *Kullback-Leibler* divergence or information gain between distributions $p_{|{\tilde{\sigma }}_{n}^{\tau -\delta }}$ and $p_{|{\tilde{\sigma }}_{n*}^{\tau -\delta }}$, defined as [51]:$$D(p_{|{\tilde{\sigma }}_{n}^{\tau -\delta }}\left|\right|p_{|{\tilde{\sigma }}_{n*}^{\tau -\delta }})=\sum_{\sigma_{n}^{\tau -\delta }}p\left(\sigma_{n}^{\tau -\delta }|{\tilde{\sigma }}_{n}^{\tau -\delta }\right)\log\frac{p(\sigma_{n}^{\tau -\delta }|{\tilde{\sigma }}_{n}^{\tau -\delta })}{p(\sigma_{n}^{\tau -\delta }|{\tilde{\sigma }}_{n*}^{\tau -\delta })}\;.$$

The Kullback-Leibler divergence is a non-negative measure $D(p_{|{\tilde{\sigma }}_{n}^{\tau -\delta }}||p_{|{\tilde{\sigma }}_{n*}^{\tau -\delta }})\geq 0$, and
$$D(p_{|{\tilde{\sigma }}_{n}^{\tau -\delta }}||p_{|{\tilde{\sigma }}_{n*}^{\tau -\delta }})=0\;,$$
only in the case of $p_{|{\tilde{\sigma }}_{n}^{\tau -\delta }}=p_{|{\tilde{\sigma }}_{n*}^{\tau -\delta }}$. In other words, as expected, if transitions are performed between equilibrium states, no contributions arise from this term. In analogy to the equilibrium Helmholtz free energy—see Equation (Equation 7)—on can define a *non-equilibrium Helmholtz free energy* of the non-equilibrium macrostate ${\tilde{\sigma }}^{t}$, $F({\tilde{\sigma }}^{t})$, as follows [43,52,53]:(21)$$F\left({\tilde{\sigma }}^{t}\right)\equiv {\langle \beta G\left(t\right)\rangle }_{{\tilde{\sigma }}^{t}}-H\left({\tilde{\sigma }}^{t}\right)\;,$$
where the average is over all the states of the macrostate ${\tilde{\sigma }}^{t}$. If one assumes that the transitions between states obey the Markov property—see Appendix A for details—one can define the increase on non-equilibrium Helmholtz free energies, δF(τ−δ,τ), as:$$\delta F\left(\tau -\delta ,\tau \right)=F\left({\tilde{\sigma }}_{n+1}^{\tau }\right)-F\left({\tilde{\sigma }}_{n}^{\tau -\delta }\right)\;,$$
where $F({\tilde{\sigma }}_{n+1}^{\tau })$ and $F({\tilde{\sigma }}_{n}^{\tau -\delta })$ are defined following Equation (Equation 21). Now, thanks to Equation (Equation 20), one arrives at:(22)$$\delta F\left(\tau -\delta ,\tau \right)=\delta D\left(\tau -\delta ,\tau \right)+\delta F_{*}\left(\tau -\delta ,\tau \right)\;,$$
being δD(τ−δ,τ) the increase in the Kullback-Leibler divergence between τ−δ and τ, namely:(23)$$\delta D\left(\tau -\delta ,\tau \right)=D(p_{|{\tilde{\sigma }}_{n+1}^{\tau }}\left|\right|p_{|{\tilde{\sigma }}_{n+1*}^{\tau }})-D(p_{|{\tilde{\sigma }}_{n}^{\tau -\delta }}\left|\right|p_{|{\tilde{\sigma }}_{n*}^{\tau -\delta }}),$$
and $\delta F_{*}\left(\tau -\delta ,\tau \right)$ the increase on Helmholtz free energy of the equilibrium macrostates associated to ${\tilde{\sigma }}_{n+1}^{\tau }$ and ${\tilde{\sigma }}_{n}^{\tau -\delta }$, respectively. The sign of δD(τ−δ,τ) and its absolute value are important to understand the extent of the dissipative role of this term. Using the log-sum inequality [51] one is lead to—see details in the Appendix A:(24)δD(τ−δ,τ)≤0.

Again, using the log-sum inequality, one can prove that the above inequality becomes an equality only if the transitions are among equilibrium states—as expected, i.e., $p_{|{\tilde{\sigma }}_{n+1}^{\tau }}=p_{|{\tilde{\sigma }}_{n+1*}^{\tau }}$ and $p_{|{\tilde{\sigma }}_{n}^{\tau -\delta }}=p_{|{\tilde{\sigma }}_{n*}^{\tau -\delta }}$. If this is not the case,
|δD(τ−δ,τ)|>0.

**Second relation**. Recognizing that Equation (Equation 9) implies that:$$\ln\frac{p(\sigma_{n+1}^{\tau }\to \sigma_{n}^{\tau -\delta })}{p(\sigma_{n}^{\tau -\delta }\to \sigma_{n+1}^{\tau })}=\delta \beta G\left(\tau -\delta ,\tau \right)\;,$$
one can rewrite 〈δG(τ−δ,τ)〉 as:(25)$$\begin{array}{c} \langle \beta \delta G\left(\tau -\delta ,\tau \right)\rangle ={\langle \log\frac{p(\sigma_{n+1}^{\tau }\to \sigma_{n}^{\tau -\delta })}{p(\sigma_{n}^{\tau -\delta }\to \sigma_{n+1}^{\tau })}\rangle }_{\Sigma [\tau -\delta ,\tau ]}\;, \end{array}$$
where the average is computed over all trajectories Σ[τ−δ,τ] from macrostate ${\tilde{\sigma }}_{n}^{\tau -\delta }$ to ${\tilde{\sigma }}_{n+1}^{\tau }$. Furthermore, markoviantity in the transition probabilities implies that:$$\begin{array}{ccc} p(\sigma_{n+1}^{\tau }|{\tilde{\sigma }}_{n+1}^{\tau -\delta }) & = & \sum_{\sigma_{n}^{\tau -\delta }}\frac{p\left(\sigma_{n}^{\tau -\delta }|{\tilde{\sigma }}_{n}^{\tau -\delta }\right)p\left(\sigma_{n}^{\tau -\delta }\to \sigma_{n+1}^{\tau }\right)}{p({\tilde{\sigma }}_{n}^{\tau -\delta }\to {\tilde{\sigma }}_{n+1}^{\tau -\delta })}\\ & = & \sum_{\sigma_{n}^{\tau -\delta }}p_{\to }^{\tau }\left(\sigma_{n}^{\tau -\delta },\sigma_{n+1}^{\tau }\right)\;. \end{array}$$

Now we develop the 〈…〉 part of Equation (Equation 25). From the definition of $p_{\to }^{\tau }\left(\sigma_{n}^{\tau -\delta },\sigma_{n+1}^{\tau }\right)$ given in Equation (Equation 12), and averaging directly, one arrives at:$$\langle \ldots \rangle =\sum_{\sigma_{n}^{\tau -\delta }}\sum_{\sigma_{n+1}^{\tau }}p_{\to }^{\tau }\left(\sigma_{n}^{\tau -\delta },\sigma_{n+1}^{\tau }\right)\log\frac{p(\sigma_{n+1}^{\tau }\to \sigma_{n}^{\tau -\delta })}{p(\sigma_{n}^{\tau -\delta }\to \sigma_{n+1}^{\tau })}\;.$$

After some algebra, and using Equation (Equation 21), one arrives to a relation involving the global forward and backwards probabilities—see Appendix A for details:(26)$$\begin{array}{c} \delta F\left(\tau -\delta ,\tau \right)=-D(p_{\to }^{\tau }\left|\right|p_{\leftarrow }^{\tau })+\log\frac{p({\tilde{\sigma }}_{n+1}^{\tau }\to {\tilde{\sigma }}_{n}^{\tau -\delta })}{p({\tilde{\sigma }}_{n}^{\tau -\delta }\to {\tilde{\sigma }}_{n+1}^{\tau })}, \end{array}$$
where $p_{\leftarrow }^{\tau }$ is the backwards probability of trajectories, defined in Equation (Equation 12). Specifically, $D(p_{\to }^{\tau }||p_{\leftarrow }^{\tau })$ reads:(27)$$\begin{array}{ccc} D(p_{\to }^{\tau }||p_{\leftarrow }^{\tau }) & = & \sum_{\sigma_{n}^{\tau -\delta }}\sum_{\sigma_{n+1}^{\tau }}p_{\to }^{\tau }\left(\sigma_{n}^{\tau -\delta },\sigma_{n+1}^{\tau }\right)\times \;\;\;\;\;\\ & & \;\;\;\;\;\;\times \log\frac{p_{\to }^{\tau }\left(\sigma_{n}^{\tau -\delta },\sigma_{n+1}^{\tau }\right)}{p_{\leftarrow }^{\tau }\left(\sigma_{n}^{\tau -\delta },\sigma_{n+1}^{\tau }\right)}\;. \end{array}$$

If one assumes that there is no dissipation in the trajectory itself, and that the transition between states is performed in a quasi-stationary way, then $D(p_{\to }^{\tau }||p_{\leftarrow }^{\tau })=0$. We highlight that this is true as long as the trajectories are balanced and no currents are present in the system. In general, one has that, due to the non-negativity of the Kullback-Leibler divergence:$$D(p_{\to }^{\tau }||p_{\leftarrow }^{\tau })\geq 0\;.$$

#### 3.3.2. Non-Equilibrium Duplication Thresholds and Work Relations

Equation (Equation 26) encodes the relation between forward and backwards duplication probabilities. Indeed, exponentiating, one arrives directly at:(28)$$\frac{p({\tilde{\sigma }}_{n}^{\tau -\delta }\to {\tilde{\sigma }}_{n+1}^{\tau })}{p({\tilde{\sigma }}_{n+1}^{\tau }\to {\tilde{\sigma }}_{n}^{\tau -\delta })}=e^{-\beta \delta F\left(\tau -\delta ,\tau \right)-D(p_{\to }^{\tau }||p_{\leftarrow }^{\tau })}\;.$$

The above relation gives us an exact relation between duplication and fusion probabilities in a general class of non-equilibrium cases. In consequence, Equation (Equation 28) provides a necessary and sufficient condition for the triggering of a duplication event after the application of the protocol ψ(t). The above equation leads to the following duplication threshold:(29)$$F\left({\tilde{\sigma }}_{n1}^{\tau }\right)<F\left({\tilde{\sigma }}_{n}^{\tau -\delta }\right)-D(p_{\to }^{\tau }\left|\right|p_{\leftarrow }^{\tau })\;.$$

If we notice, as we did in the quasi-static case, that we can impose that $\delta \langle G\rangle =\delta \langle G_{a}\rangle$, where $\delta \langle G_{a}\rangle$ is the average increase on the free energy in the aggregates due only to size and frustration, as given in Equation (Equation 17):(30)$$\beta \delta \langle G_{a}\rangle <\delta S\left(\tau -\delta ,\tau \right)+D(p_{\to }^{\tau }\left|\right|p_{\leftarrow }^{\tau })\;,$$
and δS(τ−δ,τ) refers to the entropic contributions as described in Equation (Equation 18). Equations (Equation 29) and (Equation 30) explicitly show how the tension between entropic gradients and free energy gains controls the duplication process. This provides a nice, *hands-on* example of the imbalance between entropy and free energy gains that create structure and order that biology needs to overcome in order to endure.

From the order of magnitudes analysis of Section 2.5, we can roughly estimate the numerical values involved in these inequalities in the case of ternary mixtures containing *decanoic acid anhydride*, ($C_{9}H_{19}-\mathrm{CO}-O-\mathrm{OC}-C_{9}H_{19}$), a single precursor, Y≡
*decanoic acid*, ($C_{9}H_{19}-\mathrm{COOH}$), and water. As we outlined, the amount of aggregates in one milliliter of emulsion is ~$10^{13}$. Therefore, if we consider that just before the duplication the *extra* free energy was exactly the free energy of the aggregate at the optimal packing, we have that $\delta \langle G_{a}\rangle ∼G_{a}^{★}/n$. Since, from Equation (Equation 11), we know that $G_{a}^{★}∼10^{-13}$, we have that:$$\delta \langle G_{a}\rangle ∼10^{-26}\;J/\mathrm{aggregate}\;.$$

From that, considering T=300 K, we have that $\beta ={\left(k_{B}T\right)}^{-1}∼10^{-21}{\;J}^{-1}$, where $k_{B}$ is the Boltzmann constant. Therefore, from Equation (Equation 30) one can estimate the minimum (statistical) entropy gradient as:$$\delta S\left(\tau -\delta ,\tau \right)+D(p_{\to }^{\tau }\left|\right|p_{\leftarrow }^{\tau })∼-10^{-5}\;\mathrm{nats}\;.$$

We recall that this is entropy excludes the contribution of the heat generated in non-equilibrium transitions.

We now revise the work relations in this general non-equilibrium case. The non-equilibrium work performed over the system is:(31)$$W_{\psi }=\Delta F\left(0,\tau \right)\;.$$

From the definition of work invested over the system, one can derive the minimum work invested to trigger a duplication event. Indeed, let us suppose now that we take as the final point the equilibrium macrostate ${\tilde{\sigma }}_{n+1}^{\tau_{\infty }}$, with probability distribution $p(\sigma_{n+1}^{\tau_{\infty }}|{\tilde{\sigma }}_{n+1}^{\tau_{\infty }})$. As we said in the description of the protocol, the action of ψ(t) stops at t=τ and then the systems relaxes towards an equilibrium in a quasi-static way. One can in consequence, calculate the minimal amount of required work invested over the system through the protocol to trigger a duplication event, to be named $W_{\psi }^{★}$:$$W_{\psi }^{★}=\Delta F_{★}\left(0,\tau_{\infty }\right)\;.$$

With the above relation, one can compute the amount of *dissipated* work, $W_{\psi }^{\mathrm{diss}}$, due to non-equilibrium loses:$$W_{\psi }^{\mathrm{diss}}=W_{\psi }-W_{\psi }^{★}\;.$$

In consequence, from the definition of the non-equilibrium free energy given in Equation (Equation 22), one can find an exact expression for the dissipated work:(32)$$W_{\psi }^{\mathrm{diss}}=-\int_{0}^{\tau }\frac{dD(t)}{dt}dt\;.$$

We observe that, consistently, $W_{\psi }^{\mathrm{diss}}\geq 0$, due to inequality (Equation 24). We now retake the exploration of the orders of magnitude involved in our problem by using again the example of the ternary mixture presented in Section 2.5. The free energy of a single aggregate will determine, by construction, the minimum (non-dissipated) work needed to invest into the system to trigger the formation of an aggregate. Therefore, thanks to Equation (Equation 11), we can bound numerically the order of magnitude of this work:$$W_{\psi }^{★}>G_{a}^{★}∼10^{-13}\;J\;.$$

From this, and knowing from Section 2.5 that the amount of oil vesicles is around $10^{13}$ in a milliliter of microemulsion, we can conclude, under the assumption that a linear increase of volume to accommodate new protocells does not impact dramatically in the energy landscape, that the amount of work we need to invest to duplicate the amount of protocells contained initially in a liter of emulsion is lower bounded as:$$W_{\psi }^{★}\left(1\;L\right)≳1\;\mathrm{kJ}\;.$$

Finally, we can compute the amount of entropy produced throughout the whole process, $S_{\psi }$, by collecting the entropic terms, and adding the entropy produced by the destabilization of the system after each application of the protocol, βQ(t):$$S_{\psi }=\Delta H\left(0,\tau_{\infty }\right)+\int_{0}^{\tau_{\infty }}D(p_{\to }^{t}\left|\right|p_{\leftarrow }^{t})dt+\int_{0}^{\tau }\beta Q_{\psi }\left(t\right)\;.$$

Recall, again, that $D(p_{\to }^{\tau }||p_{\leftarrow }^{\tau })\geq 0$. We remind that here we did not specify the formal shape of the last term, corresponding to the heat produced within the non-equilibrium trajectories that destabilize the system right after the application of the protocol. We warn the reader that the potential relations between the dissipated work δD(τ−δ,τ) and the entropy produced through the relaxation process, $D(p_{\to }^{\tau }||p_{\leftarrow }^{\tau })$ are not studied here, but can contain relevant information for the conditions of the duplication process. Similar relations are studied in the context of the analysis of the structure of the second law [54], thermodynamics of computation [55], and work/energy relations in coarse grained approaches [56].

### 3.4. The Perpetuation of the Duplication Process

A crucial condition for the emergence of synthetic-living entities is the capacity for perpetuating the duplication process. In the framework derived above we very briefly revise the key ingredients for this successive duplications to be maintained. Let us suppose that at time *t* the system contains n(t) aggregates in equilibrium, i.e., ${\tilde{\sigma }}_{n(t)}^{t}$. Following Equation (Equation 29), there will be a duplication if, under the action of the protocol ψ(t), if (∀n(t)∈N)(∃τ>0), the following condition holds
(33)$$F\left({\tilde{\sigma }}_{n(t)+1}^{t+\tau }\right)<F\left({\tilde{\sigma }}_{n(t)}^{t+\tau }\right)-D(p_{\to }^{t+\tau }\left|\right|p_{\leftarrow }^{t+\tau })\;.$$

If we plug Equation (Equation 30) we obtain a criteria that explicitly relates the free energy increases of the aggregates, due to frustration and size, to the entropic gradients. In that context, the duplication process will go on as long as, (∀n(t)∈N)(∃τ>0), the following condition holds: (34)$$\begin{array}{c} \beta \delta \langle G_{a}\rangle <\delta S\left(t+\tau -\delta ,t+\tau \right)+D(p_{\to }^{t+\tau }\left|\right|p_{\leftarrow }^{t+\tau })\;, \end{array}$$
where $\delta \langle G_{a}\rangle$ is the average increase of free energies of the aggregates due to size and frustration from n(t) to n(t)+1 in the time interval [t+τ−δ,t+τ] and δS(t+τ−δ,t+τ) the increase of the other entropic components, as defined in Equation (Equation 18). Inequalities (Equation 33) and (Equation 34) are the inequalities that the protocol must trigger to ensure the continuation of the duplication process. They may be called *inequalities for prevalence*. In summary, they tell us that the process can continue if the action of the protocol is able to trigger an imbalance between entropic contributions and free energy gradients favouring the equilibrium state containing n(t)+1 aggregates as: 

Δ(*aggregate free energy*) < Δ(*entropic gradients*)

In Figure 4 we described a potential trajectory of a successive duplication process. We therefore derived a specific example of the race between entropy and free energy increases that enables the perpetuation of the duplication process. Other circumstances must be taken into account. For example, the volume of the system should increase in proportion to the aggregate number, in order to keep the concentration of chemical species inside the ranges in which the system remains in the phase where aggregates are formed. A significant change of this concentrations could result into a change on the phase of the system, where the preferred structures could no longer be spherical aggregates. Other circumstances, such as the specific application of the protocol, could also interfere the duplication process.

## 4. Discussion

In this paper we explored in depth the thermodynamics of duplication thresholds in a generic emulsion system made of an arbitrary set of lipid and precursor species. This feasible, yet artificial system enables us to overcome the tremendous complexity of the duplication process in actual living entities, such as cells. The thermodynamic landscape has been carefully constructed, accounting for the contributions due to surface tension, volume of the aggregates, entropic contributions and total amount of chemical species within the systems, all summarized in the definition of the Gibbs free energy of the state, Equation (Equation 4). An abstract protocol is proposed, driving the system away from the equilibrium state and resulting, eventually, in a duplication event. We approached the problem from the equilibrium framework, assuming that the process is a succession of equilibrium states, and from a non-equilibrium perspective, where the visited states may not be equilibrium ones.

Fundamental relations involving free energies and duplication probabilities, Equation (Equation 28), duplication thresholds, Equations (Equation 29) and (Equation 30), necessary work to be invested over the system by the protocol to trigger a duplication event, Equation (Equation 31), dissipated work, Equation (Equation 32) or the conditions for the perpetuation of the duplication cycle, Equations (Equation 33) and (Equation 34) have been derived. These relations invoke the explicit energy landscape provided by the free energies and set the abstract conditions for a duplication process to be triggered and, eventually maintained. It is worth to emphasize that they show explicitly the structure of the race between entropic forces and free energy gains to generate structure and preserve it. The synthetic approach, therefore, enabled us to convey a very detailed picture of the thermodynamical tensions involved in the process of creation and perdurability of living entities.

Further explorations should target more systematically specific systems, with quantitatively testable observables. The study of specific systems should also include the conditions of feasibility, in terms of microemulsion phases, of the aggregate duplication, avoiding transitions to non-aggregate phases, possible in emulsion systems. In the same line, a rigorous exploration of the orders of magnitude involved in the abstract relations derived above would add a necessary layer towards the quantification and, eventually, empirical test of the above predictions. Complementarily, the exploration of the constraints imposed by different protocol strategies could shed light to the potential prebiotic scenarios, where possibly circadian cycles play a crucial role in creating free energy sources driving the system towards imbalance, destabilization, and duplication. In addition, more complex free energy landscapes allowing bilayer membranes, more realistic when compared to biological structures than the single layer approach used here, could refine the triggering points for duplication events to occur. In a different direction, an in depth study of the dissipation within the trajectories themselves—assumed to observe detailed balance in the above developments—would generalize the approach, making it more realistic and providing predictions on dissipated heat which could be presumable testable. Finally, the interesting relations involving dissipation and information measures could be explored to be the seed of further developments linking information and duplication processes, in line to the results exposed in [31], and, perhaps, clear the conditions for the emergence of inheritable information—thus the appearance of differentiated traits between elements of the system—intrinsically linked to the duplication process and, in the long term, trigger darwinian dynamics.

## Figures and Tables

**Figure 1 life-09-00009-f001:**
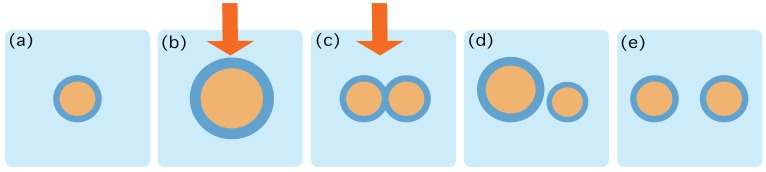
Schematic characterization of the role of the protocol ψ(t). (**a**) At time t=0 the system is in contact to a thermal reservoir at inverse temperature β, and in an equilibrium state containing 1 aggregate. (**b**,**c**) The protocol starts by increasing the number of lipids and precursors and providing energy that may trigger chemical reactions. The action of the protocol is depicted by the red arrow. This process may change the energy landscape provided by Equation (Equation 4) and thereby destabilize the structure of the aggregates, eventually creating more and more frustration in the surface. (**d**) At time τ the energy gradients favour the duplication and the protocol stops. (**e**) The system relaxes towards an equilibrium state containing 2 aggregates.

**Figure 2 life-09-00009-f002:**
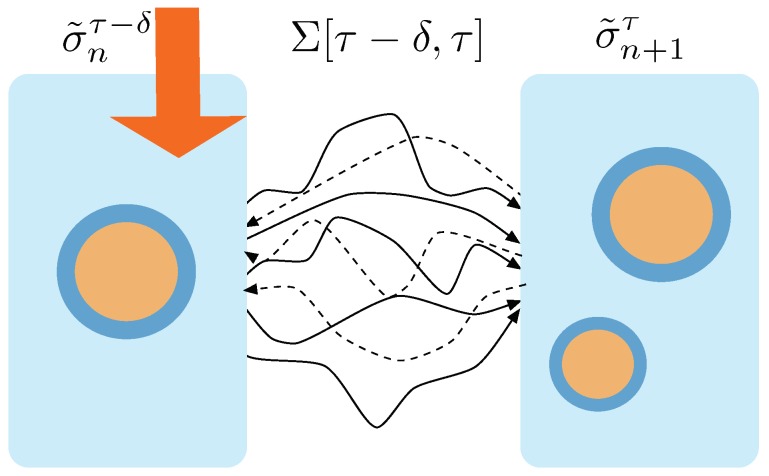
Trajectories between macrostates. At time τ−δ the macrostate containing 1 aggregate receives the action of the protocol and transits to a macrostate containing 2 aggregates. This transition can performed through any of the trajectories connecting the states that realize one or the other macrostate. The ensemble of trajectories connecting these two macrostates is called Σ[τ−δ,τ]. Forward trajectories are depicted with solid lines. Dashed lines correspond to time reversal trajectories, i.e., trajectories obtained through the protocol running under time reversal.

**Figure 3 life-09-00009-f003:**
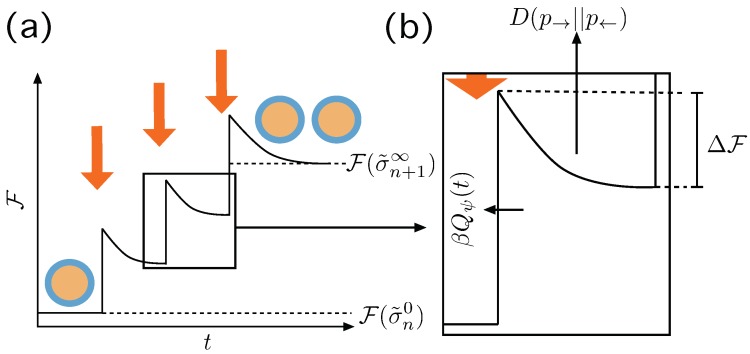
Irreversible action of the Protocol ψ. (**a**) At t=0 we have a macrostate ${\tilde{\sigma }}_{n}^{0}$ in equilibrium and the protocol induces a change in the boundary conditions that destabilizes the system eventually making it to jump to a non equilibrium state, producing an amount of entropy $\beta Q_{\psi }\left(t\right)$. Then the system relaxes—maybe not completely—until the next action of the protocol until there is a stable division event and the system relaxes completely. (**b**) Details of the transition, with the thermodynamic quantities involved. The jump experienced by the system from its previous state is $D(p_{|{\tilde{\sigma }}^{t}}||p_{|{\tilde{\sigma }}_{*}^{t}})$, and ΔF is the energy gradient that leads to the new macrostate. The entropy produced through this, possibly partial, relaxation process is $D(p_{\to }||p_{\leftarrow })$—see text.

**Figure 4 life-09-00009-f004:**
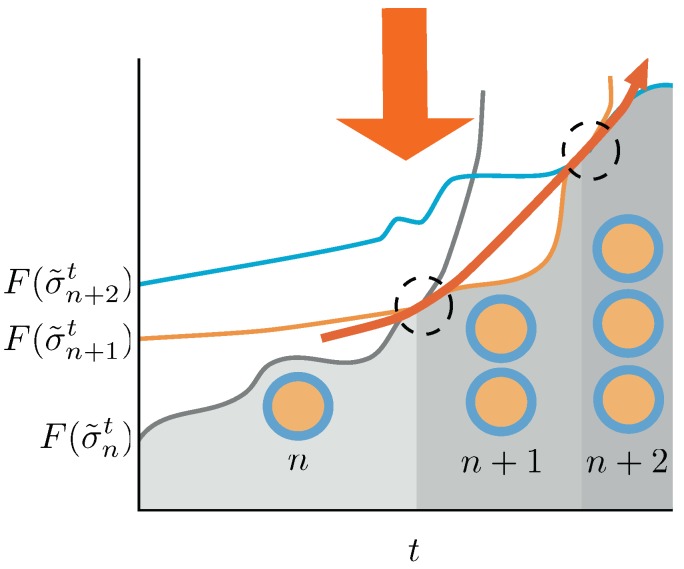
Schematic picture of the conditions for the duplication process to be sustained in time. Duplication events are indicated by dashed circles. A system with *n* aggregates—gray line—in equilibrium receives the action of the protocol changing the energy landscape. Its Helmoltz free energy increases until a point in which the Helmholtz free energy of a macrostate containing n+1 aggregates—orange line—is lower than the one for the *n* aggregates, and a duplication event occurs. If we switch on the protocol again, the system increases its Helmholtz free energy until a point in which, eventually, the Helmholtz free energy differences trigger again a duplication event—blue line. If the protocol is able to destabilize the system from n(t) to n(t)+1 aggregates for any *t*, the duplication process will continue unboundedly in time. In this figure we described a quasi-static approach, which makes use of equilibrium Helmholtz free energies for the sake of clarity. The non-equilibrium case is thoroughly discussed in the text.

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
