# Peer review of "Thermodynamics of Duplication Thresholds in Synthetic Protocell Systems"

_life, 2019, doi:10.3390/life9010009_

Round 1
Reviewer 1 Report
This paper treats the equilibrium thermodynamics of lipids based on the Gibbs equation. The thermodynamics of lipid vesicles, micelles, and sheet emulsions, from formation to growth to fission (duplication) is an important subject because of the many technological applications, such as liposomes for drug delivery. The thermodynamics of lipids has therefore been extensively studied previously and in much detail. It was disappointing that the author did not start with an introduction to the very large body of existent work on lipid equilibrium thermodynamics and then specify how their analysis and results are different from, or add to, previous works. What characterizes the present work is that it is more general than usual and the author derives a general formula which they claim amounts to a free energy threshold which can predict vesicle duplication, and this would certainly be of interest to the community. However, it is hard to judge the validity of the quasi-equilibrium derivation since no application to real lipid systems was attempted. I would have liked to see a specific application of the formalism, for example, the determination of the critical vesicle concentration (CVC) for a particular case, and comparison with experiment, in order to validate the formalism.
Where I have particular trouble with the manuscript is where the author attempts to apply this equilibrium thermodynamic formalism to situations which the author refers to as “out of equilibrium”. The author confines this analysis to a protocol which produces a perturbation of the boundary conditions and leads to abrupt changes to the Gibbs free energy surface. However, this is not a truly non-equilibrium analysis. Although duplication of lipid vesicles can occur through a variety of external perturbations, this is certainly not the whole non-equilibrium story, particularly with respect to vesicles which are part of living systems (the apparent object of this study). In living systems, vesicles are under externally imposed generalized thermodynamic potentials and this leads to reorganization (dissipative structuring) of which duplication (replication) is one resulting phenomenon. For example, non-linear autocatalytic chemical reactions dissipating an imposed chemical affinity normally gives rise to duplication of structure in living systems. This class of non-equilibrium situations has been completely ignored in the present manuscript but is, without doubt, the most important aspect required for understanding duplication of vesicles in living systems.
The work presented by the author is therefore interesting and useful for quasi-equilibrium conditions where the system is externally perturbed, but I have serious doubts if the formalism can be used as a good model for replication in any living system. This work therefore has applicability to the liposome community (for example in designing drug delivery systems) but has little relevance to replication in living systems which are structured under externally imposed generalized chemical potentials. I would therefore recommend the author to consider revising and resubmitting the manuscript taking into account the following major corrections;
Major Corrections;
1) Fisrt define clearly the classes of non-equilibrium situations that are going to be considered in the paper and distinguish clearly these situations from the non-equilibrium situations present in living organisms. Perhaps the paper should be better directed towards the liposome construction scientists, e.g. the drug delivery community, rather than the systems biology community.
2) Present a comprehensive overview of work already performed on the equilibrium thermodynamics of lipid systems and explain clearly how this work is different from, or adds to, previous works.
3) Provide a specific example of the application of the derived formalism, for example an actual calculation of the free energy input required for vesicle duplication and comparison with experiment, or the calculation of the equilibrium critical concentration required for vesicle formation (CVC) compared with experiment.
Minor Corrections;
There are a number of phrases in the introduction which are false and betray a lack of understanding of the chemistry and physics (particularly of non-equilibrium thermodynamics) involved in living systems. For example;
1) The author writes, “The perpetuation and development of life on earth involves an open race between free energy gains and entropy [1, 2].” In equilibrium thermodynamics, free energy is the sum of internal energy and a term of minus the temperature times the entropy. Minimizing the free energy therefore implies minimizing the internal energy and/or maximizing the entropy. However, this is equilibrium physics which applies only to isolated systems. This does not apply to living systems which are open systems under a generalized thermodynamic potential. In this case we only have the general evolutionary criterion of Classical Irreversible Thermodynamics which is not a potential that can be optimized to determine the direction of evolution (see Prigogine, 1961). What is driving the evolution of living systems towards greater complexity is the increase in global entropy production of system plus environment, not “free energy gains” of the individual.
2) The phrase “How living beings have been able to overcome the entropic forces to develop increasingly complex individuals which, in turn, maintain their functionality in an apparent contradiction with the laws of thermodynamics is an open question and one of the hardest questions of modern science [2{11]” is false. Classical Irreversible Thermodynamics has shown since the early 1960’s that during the evolution of an open system (living beings are open systems) towards a stationary state, the entropy of the system can decrease.
3) There are a number of problems with the English and poorly constructed sentences which should be corrected, for example, “…in the approach take here…” should read “… in the approach taken here…”, “…application of the protocol, could also interfere the duplication process” should read “…application of the protocol, could also interfere with the duplication process”, etc.
Author Response
Response to reviewer #1
I acknowledge the effort performed by the reviewer, for it is clear the her/his comments and criticisms have substantially improved the quality of the manuscript. In the new manuscript, changes can be clearly tracked because add-ons are highlighted in bold-faced font.
Reviewer comment:
1) First define clearly the classes of non-equilibrium situations that are going to be considered in the paper and distinguish clearly these situations from the non-equilibrium situations present in living organisms. Perhaps the paper should be better directed towards the liposome construction scientists, e.g. the drug delivery community, rather than the systems biology community.
Response:
In the new manuscript, there is a clear statement on where and how the approach applies. As the reviewer may see, at the beginning of the section III.C "Non-equilibrium approach" there is a clear statement on the scope of the approach.
Concerning the audience the paper should be directed to, the paper would belong to a special issue where synthetic systems mimicking living entities play a big role. The ausience will be mainly biologists and physicists. This is observation is relevant also for the next point.
Reviewer comment:
2) Present a comprehensive overview of work already performed on the equilibrium thermodynamics of lipid systems and explain clearly how this work is different from, or adds to, previous works.
Response:
As the reviewer can see in the new version, I added new citations to better locate my contribution. I cannot agree, however, with the fact that I should add an even more comprehensive citation list, because my aim is to locate the proposal within the framework of artificial/synthetic life. An in-depth excursion to the world of lipid systems, even very interesting, would deviate from the general target of the paper: To explore the physics behind a consistent, physically feasible duplication process. This must include bibliography of evolution, biochemistry, physics of colloids, and non-equilibrium statistical physics, in particular for this last field, stochastic thermodynamics, the framework I've chosen for the reasons exposed in the paper. All in all has, of course, its limitations, but the aim is to convey a way to integrate all fields to address a particular problem.
The new citations try to bridge the lack identified by the reviewer and the target of the paper:
[1] Rashevky, N. Mathematical Biophysics:Physico- Mathematical Foundations of Biology Univ. of Chicago Press:Chicago Press, 1938.
[22] Hanczyc, M. M., Toyota, T., Ikegami, T., Packard, N., and Sugawara, T. ”Fatty Acid Chemistry at the Oil- Water Interface: Self-Propelled Oil Droplets” J. Am. Chem. Soc., 129 (30), 2007:9386–9391.
[23] Mansy,S.S.,Schrum,J.P.,Krishnamurthy,M.,Tob ́e, S., Treco, D. A., and Szostak, J. W. ”Template-directed synthesis of a genetic polymer in a model protocell” Na- ture 454 2008:122–125.
[24] Toyota, T. Maru, N., Hanczyc, M. M., Ikegami, T., and Sugawara, T. ”Self-Propelled Oil Droplets Con- suming Fuel Surfactant” J. Am. Chem. Soc., 131 (14), 2009:5012–5013.
[25] Attwater, J., Wochner, A., Pinheiro, V. B., Coulson, A., and Holliger, P. ”Ice as a protocellular medium for RNA replication” Nature Communications 1, 2010:76.
[27] Corominas-Murtra, B., Fellermann, H., and Sole, R. ”Protocell cycles as thermodynamic cycles” in The inter- play between thermodynamics and computation in natural and artificial systems, Wolpert, D. H., Kempes, C., Grochow, J. A., and Stadler P. F. (eds) Santa Fe Institute press: Santa Fe, NM (accepted, to appear).
[31] Taylor, J. W., Eghtesadi, S. A., Points, L. J., Liu T., and Cronin, L. ”Autonomous model protocell division driven by molecular replication” em Nature Communications 8 (1) 2017:237
[47] Kaganer,V.M.,Mohwald,H.,and Dutta,”Structure and phase transitions in Langmuir monolayers” P. Rev. Mod. Phys. 71, 1999:779 .
[48] Caschera, P., Rasmussen, S. and Hanczyc, M. M. ”An Oil Droplet Division-Fusion Cycle” ChemPlusChem DOI: 10.1002/cplu.201200275.
Reviewer comment:
3) Provide a specific example of the application of the derived formalism, for example an actual calculation of the free energy input required for vesicle duplication and comparison with experiment, or the calculation of the equilibrium critical concentration required for vesicle formation (CVC) compared with experiment.
Response:
A totally new section II.E "Orders of magnitude" has been added providing quantitative results of a particular realization of the framework presented. Also in the section III.C "Non-equilibrium approach" some numerical computations are provided, to explore the orders of magnitude of the system works with.
Reviewer comment:
There are a number of phrases in the introduction which are false and betray a lack of understanding of the chemistry and physics (particularly of non-equilibrium thermodynamics) involved in living systems. For example;
1) The author writes, “The perpetuation and development of life on earth involves an open race between free energy gains and entropy [1, 2].” In equilibrium thermodynamics, free energy is the sum of internal energy and a term of minus the temperature times the entropy. Minimizing the free energy therefore implies minimizing the internal energy and/or maximizing the entropy. However, this is equilibrium physics which applies only to isolated systems. This does not apply to living systems which are open systems under a generalized thermodynamic potential. In this case we only have the general evolutionary criterion of Classical Irreversible Thermodynamics which is not a potential that can be optimized to determine the direction of evolution (see Prigogine, 1961). What is driving the evolution of living systems towards greater complexity is the increase in global entropy production of system plus environment, not “free energy gains” of the individual.
Response:
Thanks for the clarification. I removed the problematic claims.
Reviewer comment:
2) The phrase “How living beings have been able to overcome the entropic forces to develop increasingly complex individuals which, in turn, maintain their functionality in an apparent contradiction with the laws of thermodynamics is an open question and one of the hardest questions of modern science [2{11]” is false. Classical Irreversible Thermodynamics has shown since the early 1960’s that during the evolution of an open system (living beings are open systems) towards a stationary state, the entropy of the system can decrease.
Response:
Thanks for the clarification. I removed the problematic claims.
Reviewer comment:
3) There are a number of problems with the English and poorly constructed sentences which should be corrected, for example, “…in the approach take here…” should read “… in the approach taken here…”, “…application of the protocol, could also interfere the duplication process” should read “…application of the protocol, could also interfere with the duplication process”, etc.
Response:
Thanks for raising this point. I corrected the grammar throughout the text.
Reviewer 2 Report
BRIEF DESCRIPTION OF THE WORK
In this work the author studies a synthetic system made of small synthetic protocell aggregates. In particular, he analyses the thermodynamics of duplication thresholds in a generic emulsion system made of an arbitrary set of lipid and precursor species. For this purpose, the authors introduced a protocol that changes the energy landscape, without separating the system from the heath bath neither changing the whole system’s volume. The main result of the work is to have found inequalities that the protocol must trigger to ensure the continuation of the duplication process [Refs. Eqs (30) - (31) and Fig. (4) in the manuscript]. In particular, the process will continue as long as Eq. (31) is satisfies. Hence, thanks to this “artificial” (but feasible) system we understand complex phenomena like the duplication process in living cells.
GENERAL CONSIDERATIONS
Personally, I appreciate this work: it is a smart attempt to treat complex phenomena, such as the duplication process in living cells, by means of a relative simple model (the author calls it “artificial system”).
From the thermodynamic point of view, I did not detect any contradiction with the basic laws governing thermodynamics (and Non-Equilibrium Statistical Thermodynamics).
QUESTIONS/SUGGESTIONS
However, I would like to address the following points to the author.
Q1) Pictures 1)-3) in the manuscript illustrate quite well the definition, the role of the and the action of the protocol ψ. In addition, Fig. 4) depicts well a trajectory of a successive duplication process to be sustained in time. However, in this work there is a lacking of quantitative results. In general, when it is proposed a model (or an “artificial system”) it is also customary to produce numerical results (with graphics) just to provide, at least, the order of magnitude of the relevant variables entering in the problem. For example, according to the model, what are the range of values of the Gibbs free energy given by Eq. (3) ? Again, may we quantify, maybe by citing a concrete case, the fundamental relation given by Eq. (31) ?
Q2) The authors defines the non-equilibrium Helmholtz free energy of the non-equilibrium macro-state [σ ̃t, F(σ ̃t)] by Eq. (18) [in analogy to the equilibrium Helmholtz free energy given by Eq. (6)]. This is certainly correct, and I have non problem about it. However, the validity of Eq. (18) must be consistent with the validity with Eq. (3) that, as known, rests upon the validity of the local equilibrium principle. May the author add some supplementary sentences in order to clarify this important aspect from the thermodynamic point of view ?
Q3) The author found the thermodynamic relation between the free energy and entropic gradients for the occurrence and prevalence of duplication phenomena. He also established the (thermodynamic) relation between work and energy by determining, in this way, the work to be accomplished by the protocol on the system, which is necessary to trigger a duplication event. These results are very interesting and, in my opinion, they should be explained more in detailed from the thermodynamic point of view (to this purpose, please consider also Fig. (2) in the manuscript. Please, try to interpret this also from macroscopic thermodynamics). This will certainly attract specialists in the Thermodynamics of Irreversible Processes working on duplication process in living cells.
CONCLUSIONS
This work is a meaningful attempt to treat complex systems like the duplication of living cells (and successively, why not, "duplication of carcinogenic cells"). Hence, in my opinion, undoubtedly, it deserves to be published. I think that the author may (strongly) attract the interest of the community working on duplication in living cells (through the tools of the Thermodynamics of Irreversible Processes and Non-equilibrium Statistical Thermodynamics) by answering to the above-mentioned questions Q1)-Q3) (in particular, question Q3)].
Author Response
Response to reviewer #1
I acknowledge the effort performed by the reviewer, for it is clear the her/his comments and criticisms have substantially improved the quality of the manuscript. In the new manuscript, changes can be clearly tracked because add-ons are highlighted in bold-faced font.
Reviewer's comment:
Q1) Pictures 1)-3) in the manuscript illustrate quite well the definition, the role of the and the action of the protocol ψ. In addition, Fig. 4) depicts well a trajectory of a successive duplication process to be sustained in time. However, in this work there is a lacking of quantitative results. In general, when it is proposed a model (or an “artificial system”) it is also customary to produce numerical results (with graphics) just to provide, at least, the order of magnitude of the relevant variables entering in the problem. For example, according to the model, what are the range of values of the Gibbs free energy given by Eq. (3) ? Again, may we quantify, maybe by citing a concrete case, the fundamental relation given by Eq. (31) ?
Response:
A totally new section II.E "Orders of magnitude" has been added providing quantitative results of a particular realization of the framework presented. Also in the section III.C "Non-equilibrium approach" some numerical computations are provided, to explore the orders of magnitude of the system works with. In addition, I added citations to better locate the manuscript in the field. Specifically:
[1] Rashevsky, N. Mathematical Biophysics:Physico- Mathematical Foundations of Biology Univ. of Chicago Press:Chicago Press, 1938.
[22] Hanczyc, M. M., Toyota, T., Ikegami, T., Packard, N., and Sugawara, T. ”Fatty Acid Chemistry at the Oil- Water Interface: Self-Propelled Oil Droplets” J. Am. Chem. Soc., 129 (30), 2007:9386–9391.
[23] Mansy,S.S.,Schrum,J.P.,Krishnamurthy,M.,Tob ́e, S., Treco, D. A., and Szostak, J. W. ”Template-directed synthesis of a genetic polymer in a model protocell” Na- ture 454 2008:122–125.
[24] Toyota, T. Maru, N., Hanczyc, M. M., Ikegami, T., and Sugawara, T. ”Self-Propelled Oil Droplets Con- suming Fuel Surfactant” J. Am. Chem. Soc., 131 (14), 2009:5012–5013.
[25] Attwater, J., Wochner, A., Pinheiro, V. B., Coulson, A., and Holliger, P. ”Ice as a protocellular medium for RNA replication” Nature Communications 1, 2010:76.
[27] Corominas-Murtra, B., Fellermann, H., and Sole, R. ”Protocell cycles as thermodynamic cycles” in The inter- play between thermodynamics and computation in natural and artificial systems, Wolpert, D. H., Kempes, C., Grochow, J. A., and Stadler P. F. (eds) Santa Fe Institute press: Santa Fe, NM (accepted, to appear).
[31] Taylor, J. W., Eghtesadi, S. A., Points, L. J., Liu T., and Cronin, L. ”Autonomous model protocell division driven by molecular replication” em Nature Communications 8 (1) 2017:237
[47] Kaganer,V.M.,Mohwald,H.,and Dutta,”Structure and phase transitions in Langmuir monolayers” P. Rev. Mod. Phys. 71, 1999:779 .
[48] Caschera, P., Rasmussen, S. and Hanczyc, M. M. ”An Oil Droplet Division-Fusion Cycle” ChemPlusChem DOI: 10.1002/cplu.201200275.
Reviewer's comment
Q2) The authors defines the non-equilibrium Helmholtz free energy of the non-equilibrium macro-state [σ ̃t, F(σ ̃t)] by Eq. (18) [in analogy to the equilibrium Helmholtz free energy given by Eq. (6)]. This is certainly correct, and I have non problem about it. However, the validity of Eq. (18) must be consistent with the validity with Eq. (3) that, as known, rests upon the validity of the local equilibrium principle. May the author add some supplementary sentences in order to clarify this important aspect from the thermodynamic point of view ?
Response:
In the new manuscript, there is a clear statement on where and how the approach applies. As the reviewer may see, at the beginning of the section III.C "Non-equilibrium approach" there is a clear statement on the scope of the approach.
Q3) The author found the thermodynamic relation between the free energy and entropic gradients for the occurrence and prevalence of duplication phenomena. He also established the (thermodynamic) relation between work and energy by determining, in this way, the work to be accomplished by the protocol on the system, which is necessary to trigger a duplication event. These results are very interesting and, in my opinion, they should be explained more in detailed from the thermodynamic point of view (to this purpose, please consider also Fig. (2) in the manuscript. Please, try to interpret this also from macroscopic thermodynamics). This will certainly attract specialists in the Thermodynamics of Irreversible Processes working on duplication process in living cells.
Response:
I don't know if I properly understand the requirement of the reviewer. I added some clarifying sentences to clearly state what is the scope and the consequences of the approach taken. In any case, I hope that the detailed quantitative example can fill the gaps found in that respect in the previous manuscript.
Round 2
Reviewer 1 Report
The author has taken into account most of my initial concerns and
criticisms of the paper and has included a new section with a specific
example. Incorrect statements regarding the thermodynamics of
irreversible systems have been removed in the revised version. Although
the author extended the first paragraph of section C to detail some of
the basic assumptions of the formalism, the identification of the regime
of applicability of the formalism is still not well defined and I leave
it up to the author to improve this before the paper is published. I
would suggest renaming section E to “Example of Free Energies of
Aggregates of Decanoic Acid”. Also, the first paragraph of this new
section should describe in detail the example. At present it is poorly
constructed.
Author Response
Response to reviewer #1
I want to acknowledge the effort performed by the reviewer, for it is clear the her/his comments and criticisms have substantially improved the quality of the manuscript. Below, I detail my responses.
Reviewer’s comment:
Although the author extended the first paragraph of section C to detail some of the basic assumptions of the formalism, the identification of the regime of applicability of the formalism is still not well defined and I leave it up to the author to improve this before the paper is published. I would suggest renaming section E to “Example of Free Energies of Aggregates of Decanoic Acid”. Also, the first paragraph of this new section should describe in detail the example. At present it is poorly constructed.
Response:
After a careful revision of the manuscript I decided not to perform substantial changes. The major change concerns to section E where I changed the name to a more specific one, as suggested by the reviewer. Essentially, I left the sections as they were in the revised version. The reason is the following: The aim of the paper is to be a self contained piece where the duplication phenomenon is dealt from the physics, and, enlarging the sections more than they are now would, in my opinion, make the paper tedious and too long. Recall that, in the current version, there are plenty of disclaimers and the degree of applicability, as well as the construction of the framework, is recurrently discussed.
Some typos and misspellings have been corrected.
I acknowledge the effort made by the reviewer and I would like to thank again the critical perspective over my manuscript.